# Educational Videos as an Adjunct Learning Tool in Pre-Clinical Operative Dentistry—A Randomized Control Trial

**DOI:** 10.3390/healthcare10020178

**Published:** 2022-01-18

**Authors:** Osama Khattak, Kiran Kumar Ganji, Azhar Iqbal, Meshal Alonazi, Hmoud Algarni, Thani Alsharari

**Affiliations:** 1Department of Operative Dentistry & Endodontics, Jouf University, Sakaka 72345, Saudi Arabia; dr.osama.khattak@jodent.org (O.K.); dr.azhar.iqbal@jodent.org (A.I.); dr.meshal.alonazi@jodent.org (M.A.); dr.hmoud.algarni@jodent.org (H.A.); 2Department of Preventive Dentistry, Jouf University, Sakaka 72345, Saudi Arabia; 3Restorative and Dental Materials Department, Faculty of Dentistry, Taif University, Taif 26571, Saudi Arabia; Thani.Alsharari@gmail.com

**Keywords:** operative pre-clinical, competency-based education, e-learning

## Abstract

Background: E-learning is an important adjunct used for teaching clinical skills in medicine dentistry. This study evaluated and compared the effectiveness of e-learning resources as an additional teaching aid to traditional teaching methods in male and female students and based on CGPA scores in a pre-clinical operative skill course. Methods: A randomized control trial was conducted in the College of Dentistry, Jouf University, to assess the impact of e-learning resources in learning clinical skills in a pre-clinical operative dentistry course. Fifty second-year dental students were randomly divided into two groups, with 25 students each. Group A (control group) was taught using traditional teaching methods, and Group B (intervention group) used e-learning resources along with traditional methods. Both groups were assessed using objective structured clinical examinations (OSCEs). Standardized forms prepared by faculty members were used to assess the students. The students also filled in a questionnaire afterwards to provide feedback regarding the e-learning resources. Results: The difference between both groups was statistically significant (*p* < 0.05). Female students performed better in three OSCE stations out of six. Furthermore, the students positively responded to the use of additional resources. Conclusion: The use of e-learning resources in pre-clinical operative dentistry courses can be a useful adjunct to traditional teaching methods and can result in better learning of dental pre-clinical operative skills.

## 1. Introduction

There has been a paradigm shift in medical education in the last decade, with the incorporation of e-learning resources in teaching and learning [1,2]. E-learning entails teaching with the aid of electronic resources. Traditional teaching involves classroom teaching that includes monitoring. With the increasing use of computers and the Internet, e-learning has become common. The Bandura social learning theory emphasizes the dynamics of the learning process, particularly for self-confidence and learning from observation of one’s own or others’ failures and the success resulting from the acquisition of new abilities. Listening to podcasts, learning from viewing videos, and others are all part of the Bandura social learning concept [3]. Dental education is also undergoing a paradigm shift in learning where various dentistry fields are witnessing innovation in teaching and learning methods. Using e-learning resources has positively impacted students in terms of understanding the basic concepts of dentistry and their application in clinical scenarios [4]. However, students view e-learning as a helpful supplement to traditional teaching methods rather than a replacement for traditional teaching methods [5].

The outcome of systematic studies on selection techniques in medical education backed up the idea that past academic achievement is a predictor of success [6]. Even though overall cumulative grade point averages (CGPA) and CGPA in science are the strongest indicators of success in dentistry schools, these may not always represent a dental student’s success in terms of clinical performance on regional tests [7].

As more importance is assigned to the impact of gender in academic achievement, research in this area is becoming more vital. As far as theoretical and practical examinations in dentistry are concerned, data show that female medical students perform better than their male classmates [8,9]. The study by Nuzhat et al. [10] in Saudi Arabia found gender-based differences in learning style preferences and the corresponding consequences on medical students’ academic performance. This study emphasized that females have more diverse preferences than male students [10]. Furthermore, female Jordanian dental students surpass males in dentistry courses, according to the reports of Sawair [11].

Clinical competency is an individual’s ability to work independently and without supervision in clinical practice. The ultimate objective of competency-based education is to enable dental students to be competent enough in the management of dental diseases. Therefore, the dental faculty must assess the same before students move from pre-clinical simulation to clinical courses. This process involves a series of progressive stages that begin with theory, then move toward pre-clinical simulation, and end at the clinical stage. Pre-clinical simulation courses provide a safe environment for students for learning clinical skills before moving on to the clinical stage [12]. Simulation labs provide an opportunity for the students to learn such skills. Operative dentistry is an extensive subject. It requires learning clinical skills, which begins from the second year of a Bachelor of Dental and Oral Surgery program (BDS). Operative dentistry skill is one of the basic skill courses in the undergraduate dental curriculum. It covers important topics, including knowledge about dental materials and their clinical application, various infection control measures for the dental unit, and the use of equipment related to operative dentistry. Students are expected to be competent in the skills mentioned above at the end of the course. It provides an opportunity to learn the basic skills of operative dentistry in pre-clinical simulation labs before proceeding to clinical courses. The objective structured clinical examination (OSCE) is an efficient method of assessing students’ clinical competency and skills [13]. It is used routinely in dentistry for student assessment [14]. Studies have shown that students feel less confident about the skills they learn using traditional, didactic teaching methods, and this makes learning difficult [15]. An opportunity to start learning about teledentistry and virtual patient management is provided during the COVID 19 epidemic [16]. Blended learning, which involves traditional teaching along with electronic resources, has shown that it can transform traditional teaching experiences into technology-enhanced learning ones, which students today find beneficial [17]. Haptic technology, such as robotics, is also gaining traction, since it allows for two-way communication between the user and the environment, allowing for a more accurate simulation of the clinical setting for learning reasons. Current literature lacked information about use of e-learning tools as a additional teaching aid for the pre-clinical operative skills of dentistry.

This study aimed at:Evaluating and comparing the effectiveness of e-learning resources as an additional teaching aid to traditional teaching methods in male and female students as well as based on the CGPA scores in a pre-clinical operative skill course.Correlating the effectiveness of e-learning resources with CGPA scores of dental students.

## 2. Materials and Methods

This randomized control trial (Figure 1) was conducted at College of Dentistry, Jouf University, KSA, from 15 January to 28 February 2021. Ethical approval was obtained by the Local Committee of Bioethics, Jouf University. The study sample comprised all second-year dental students of the 2020–2021 Bachelor of Oral and Dental Surgery program who consented (*n* = 50) to the census technique. This is a method where all members of a population are analyzed. Students who did not volunteer were excluded. Recruitment was conducted by announcements through e-mail. Informed consent was obtained from students. They were randomly distributed into two groups: control and intervention. Randomization was undertaken using a computer-generated random number for the roll numbers as per their attendance list (Figure 1).

The skill lab was scheduled as per the calendar of the curriculum for teaching the operative dentistry skills using the traditional method. The traditional method included the demonstration of matrix band placement, dental dam application, application of retraction cord, mixing of glass ionomer restorative material, operator positioning for specific tooth and disinfection of dental unit using a simulation process. The intervention group (Group B) received an e-learning resource in the form of audiovisual aids such as videos in addition to traditional demonstration methods whereas the control group (Group A) were taught by the traditional demonstration method only. The e-learning resources comprised educational videos of the same skill procedures which were already taught employing traditional methods. These resources were selected by other faculty members of the department of operative dentistry who were not involved in the study process. The total duration of the videos was around 24 min. The videos were shared only with the intervention group (Group B) through the institutional e-learning app called Blackboard^®^, with statistics tracking enabled to confirm that students viewed this video for the minimum number of five views at their convenience. According to the statistics, a reminder email was issued to students who had not watched the videos. At the end of the videos, a quiz was administered to assess their understanding by using the adaptive release feature of Blackboard in which a student can take up the quiz if and only upon they had completed the minimum number of views. After 1 week, both the groups were assessed using OSCE (Figure 2), the standard method followed in College of Dentistry, Jouf University. For the examiners, the students of both groups were anonymized. To maintain a high level of objectivity, it was ensured that the examiners were not familiar with the students from previous academic courses. The students were assessed by two examiners to maintain inter-examiner reliability using checklists for each station. The average score of the two examiners was considered the score for that station. It was ensured that the examiners did not share the results. The average scores for each station were calculated, which did not contribute to the real grades of the students, as these OSCE exams were conducted during revision sessions before summative exams.

Thereafter, feedback was obtained using a Likert scale rating (1—strongly agree to 5—strongly disagree) from the students regarding the importance of e-learning resources. The CGPA of all the students were categorized into three categories: low CGPA (<3), average CGPA (3–4), high CGPA (>4). The CGPA scores and gender information were used as dependent variables; OSCE scores were used as the outcome variable. The data were entered in Microsoft Excel and statistics were performed using Statistical Package for the Social Sciences (SPSS) IBM Corp. released 2017, version 25.0 Armonk, NY, USA. Results were presented as mean values with standard deviation. The test of significance for mean OSCE scores between the two groups and gender was assessed using Student’s *t*-test. One way ANOVA was used to assess the test of significance for mean OSCE scores in 3 different CGPA groups. The mean OSCE scores of the two groups were compared using Student’s *t*-test, and mean CGPA scores (low, average, and high) were compared using the ANOVA test. Pearson correlation test was used to correlate the OSCE scores with CGPA scores. The *p*-value of <0.05 was considered statistically significant, with 95% confidence intervals.

## 3. Results

Fifty second-year BDS students participated in the study: 33 male and 17 female students. Twenty-five students were allocated to each group, with 18 male and 7 female students in Group A and 15 male and 10 female students in Group B. This unequal distribution was due to randomization. The average score for all the students in both groups was 40.22, with a standard deviation of 4.46. The cumulative grade point average (CGPA) of the students in each group was considered. An independent-sample *t*-test was conducted to evaluate the homogeneity of the CGPA scores in both groups. There was no significant difference in the CGPA scores for Group A (M = 3.9, SD = 0.68) and Group B (M = 3.8, SD = 0.25), *p* = 0.24 (Table 1). These results suggest that both the groups were homogenous and identical concerning CGPA.

Table 1 shows the performance of students of each group at each of the six OSCE stations. All students passed, with the cut-off score set at 60%. Group B performed better at each station compared to Group A. For all six stations, the average score of the intervention group was higher than the control group. The female students performed significantly better at stations one, three, and five compared to the male students (Table 2), whereas there was no significant difference in the performance of male and female students at other stations. The three categories of students which were made based on CGPA (low, average and high) showed a significant difference in performance across all OSCE stations when one-way ANOVA was conducted (Table 3). This result also supports our hypothesis that students with better CGPA perform better in terms of skills.

A Pearson correlation coefficient was computed to assess the relationship between the CGPA score and individual station (Table 4). Overall, there was a positive correlation between them. An increase in stations scores were correlated with the increase in CGPA. There was a strong correlation (*r* = 0.701) with respect to station three, whereas moderate correlation was found with respect to stations one (*r* = 0.563), two (*r* = 0.525), four (*r* = 0.509), and five (*r* = 0.620) respectively. However, the correlation was weak with respect to station six (*r* = 0.492, *p* < 0.00). The response rate was 100% in the survey filled by the participating students who provided overall positive feedback regarding the use of additional aids.

## 4. Discussion

E-learning resources were helpful in enhancing dental students’ understanding of fundamental concepts and their application to clinical scenarios [4]. We also believe that these will also help students engage in active learning with the faculty in the skill labs, enabling more discussion on the topic.

The results obtained from our study encourage us to believe that the students can perform better when they are taught using traditional methods along with additional e-learning resources. The students in Group B who were provided with the e-learning resources performed significantly better than the other group. The results of our study are similar to those of Qutieshat et al. [18] who found that the students who studied using the hybrid model (including traditional as well as e-learning methods) performed better. E-learning resources have been useful in enhancing dental students’ understanding of fundamental concepts and their application to clinical scenarios [4]. We also believe that these will also help students engage in active learning with the faculty in the skill labs, enabling more discussion on the topic. They also found that the student’s perception of the hybrid learning method was positive, similar to our study. The positive feedback received from our sample regarding the use of videos to teach skills is similar to a study by Jang et al. who found that students’ overall perception regarding the use of videos was positive. They recommended the faculty use such resources [19]. The positive feedback emphasizes the importance of freely-available e-learning resources in teaching skills in operative dentistry. The students were able to access the videos whenever they wanted to. This helped them consolidate their knowledge through repetition. Unlike the case with traditional methods, they could watch the faculty demonstrate the skills more than once. Similar results have been yielded with the use of online videos in a pre-clinical prosthodontics course at a dental school in Germany [20]. Pre-clinical prosthodontics provides a set of skills very similar to pre-clinical operative dentistry courses, and hence their results provide external validity.

The students with better CGPA performed better than those with lower CGPA. This was expected because the students with better academic performance tend to be more academically groomed. Sound knowledge about the materials, equipment, and practical techniques forms the basis for good practical skills. They use multiple preparation techniques for their assessments and perform better. Previous studies have shown similar results in undergraduate medical education, with CGPA being a predictor of students’ performance [21]. Our study showed that the female students performed better than the male students. This is similar to studies conducted in undergraduate dental education where female students’ cumulative grades were significantly better than male students [11]. However, our results differ from a study on the effect of gender on performance in medicine in a medical college in Saudi Arabia [22], where the females performed only slightly better than their male counterparts. We believe that female students with better didactic knowledge perform better than males in dental skills, as shown in the results of our study. Our findings might potentially be utilized to garner more interest in studying gender-based differences in ability in practical courses taken by dentistry students. More exploratory studies are needed to relate gender to the performance of dental students.

We found a strong to moderate correlation between the CGPA and students’ performance at each station. However, station six revealed a weak correlation. This station involved cleaning and disinfection of the workstation. The possible explanation is that this was the only component that was not taught in the lectures, and only a demonstration of the skill was provided. This is because the students had completed the knowledge and understanding part of this aspect in another course in the previous year. Knowledge about all other stations was provided in the lecture component of the skill course. This underlines the fact that only demonstrations or the use of videos are not sufficient to learn a skill. For competence, sound theoretical background, as well as clinical demonstration, is important. A major limitation of our study is the involvement of only one cohort of students. Our study does not correlate the impact of e-learning resources on the enhancement of clinical skills by prospective follow-up. More studies involving a larger group of students in different skill-related courses of dentistry are needed to conclude this assumption.

## 5. Conclusions

Students enrolled in in pre-clinical operative dentistry skills training performed significantly better when provided with additional e-learning tools than those who were not. Thus, resources that are easily accessible to students help them perform better in the assessments. We recommend that the additional e-learning resources should be part of the teaching methodology in the skill courses of dentistry colleges. This would help students become more competent in skill courses.

## Figures and Tables

**Figure 1 healthcare-10-00178-f001:**
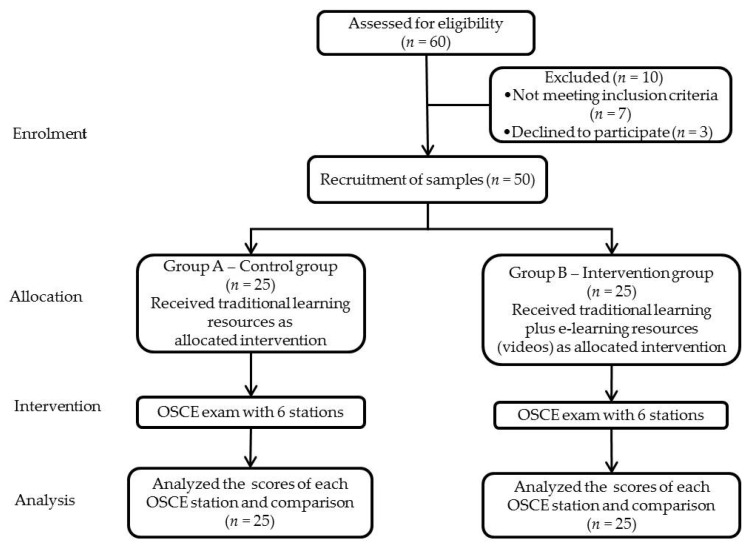
Consort chart for randomization, intervention and follow-up of study samples.

**Figure 2 healthcare-10-00178-f002:**
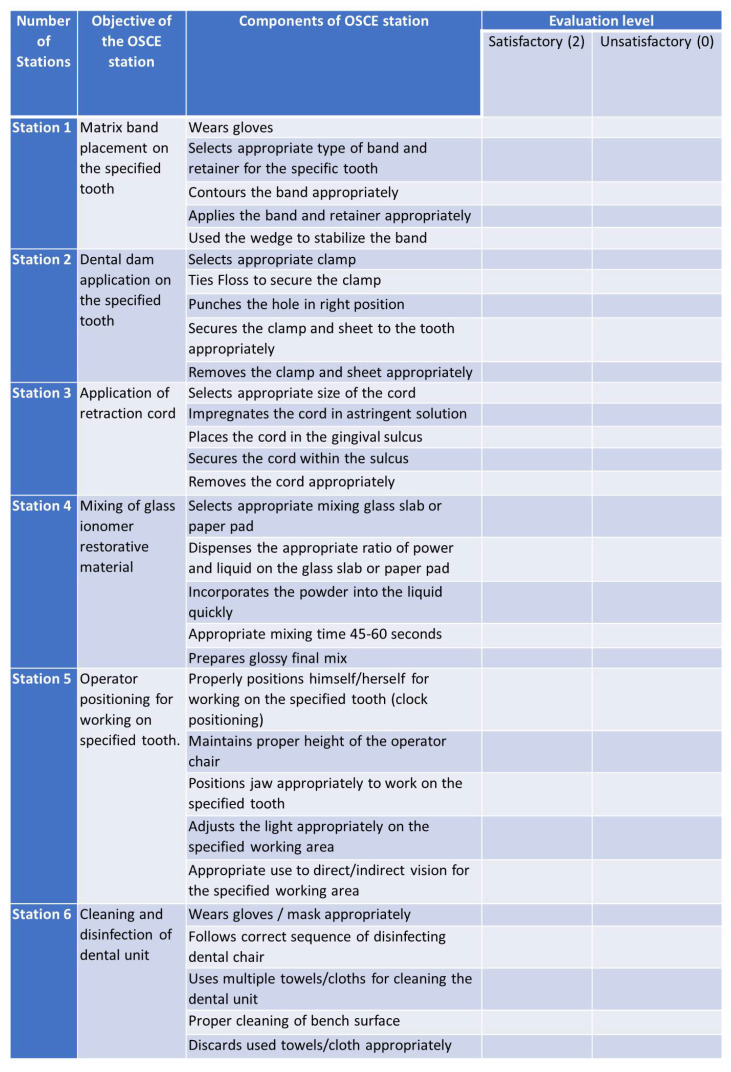
Sample description of objective structured clinical examination (OSCE) stations.

**Table 1 healthcare-10-00178-t001:** Comparison of OSCE station scores with respect to Group A and Group B.

	Group	*n*	Mean	Std. Deviation	Std. Error Mean	F Value	*p* Value
Station#1	A	25	6.480	1.045	0.209	0.763	0.067 *
	B	25	7.000	0.912	0.182		
Station#2	A	25	6.200	0.500	0.100	3.903	0.000 *
	B	25	7.000	0.866	0.173		
Station#3	A	25	6.240	0.925	0.185	0.480	0.002 *
	B	25	7.040	0.840	0.168		
Station#4	A	25	6.480	0.822	0.164	0.071	0.025 *
	B	25	7.040	0.888	0.177		
Station#5	A	25	6.280	0.791	0.158	1.186	0.000 *
	B	25	7.280	0.936	0.187		
Station#6	A	25	6.320	0.627	0.125	2.790	0.002 *
	B	25	7.080	0.996	0.199		
CGPA	A	25	3.926	0.684	0.136	28.98	0.245
	B	25	3.834	0.255	0.051		

* Statistically significant (*p* < 0.05).

**Table 2 healthcare-10-00178-t002:** Comparison of OSCE station scores with respect to gender (male and female).

	Gender	*n*	Mean	Std. Deviation	Std. Error Mean	F Value	*p* Value
Station#1	Male	33	6.424	0.902	0.157	0.763	0.001 *
	Female	17	7.352	0.931	0.225		
Station#2	Male	33	6.484	0.833	0.145	3.903	0.163
	Female	17	6.823	0.727	0.176		
Station#3	Male	33	6.424	0.902	0.157	0.480	0.026 *
	Female	17	7.058	0.966	0.234		
Station#4	Male	33	6.666	0.924	0.160	0.071	0.308
	Female	17	6.941	0.826	0.200		
Station#5	Male	33	6.515	0.972	0.169	1.186	0.007 *
	Female	17	7.294	0.848	0.205		
Station#6	Male	33	6.697	1.045	0.181	2.790	0.974
	Female	17	6.705	0.587	0.142		

* Statistically significant (*p* < 0.05).

**Table 3 healthcare-10-00178-t003:** Comparison of OSCE stations scores with respect to cumulative grade point averages (CGPA) groups (low, average and high).

OSCE	CGPA Score	*n*	Mean	Std. Deviation	Std. Error	95% Confidence Interval for Mean		*p* Value
						Lower Bound	Upper Bound	
Station#1	Low	5	5.400	0.547	0.244	4.719	6.080	0.000
	Average	11	6.272	0.646	0.194	5.838	6.707	
	High	34	7.088	0.933	0.160	6.762	7.413	
Station#2	Low	5	5.800	0.447	0.200	5.244	6.355	0.002
	Average	11	6.181	0.404	0.121	5.910	6.453	
	High	34	6.852	0.821	0.140	6.566	7.139	
Station#3	Low	5	5.200	0.447	0.200	4.644	5.755	0.000
	Average	11	6.181	0.603	0.181	5.776	6.586	
	High	34	7.000	0.852	0.146	6.702	7.297	
Station#4	Low	5	5.800	1.095	0.489	4.439	7.160	0.006
	Average	11	6.454	0.687	0.207	5.992	6.916	
	High	34	7.0000	0.816	0.140	6.715	7.284	
Station#5	Low	5	5.6000	0.547	0.244	4.919	6.280	0.000
	Average	11	6.272	0.646	0.194	5.838	6.707	
	High	34	7.117	0.945	0.162	6.787	7.447	
Station#6	Low	5	5.800	0.447	0.200	5.244	6.355	0.004
	Average	11	6.272	0.467	0.140	5.958	6.586	
	High	34	6.970	0.936	0.160	6.643	7.297	

**Table 4 healthcare-10-00178-t004:** Correlation matrix between OSCE station scores and CGPA.

	Station#1	Station#2	Station#3	Station#4	Station#5	Station#6	CGPA
Station#1	1						
Station#2	0.647 **	1					
Station#3	0.722 **	0.493 **	1				
Station#4	0.542 **	0.401 **	0.466 **	1			
Station#5	0.655 **	0.700 **	0.532 **	0.490 **	1		
Station#6	0.605 **	0.778 **	0.573 **	0.362 **	0.534 **	1	
CGPA	0.563 **	0.525 **	0.701 **	0.509 **	0.620 **	0.492 **	1

** Statistically significant (*p* < 0.01).

## Data Availability

Data will be made available on request.

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
