# Peer review of "Educational Videos as an Adjunct Learning Tool in Pre-Clinical Operative Dentistry—A Randomized Control Trial"

_healthcare, 2022, doi:10.3390/healthcare10020178_

Round 1

Reviewer 1 Report

abstract

50 second-year dental students were randomly divided in 2 groups. ->good for statistical reasons, but students talk to each other, so E-learning could also be used by students in other groups

better test one group of second-year students in 2021 (traditional) and then again in 2022(e-learning)/ multicentre study

group A: what are these traditional teaching methods (there are also differences, there is not only one traditional teaching method)

group B: is the intervenmtion group: but it is clear, that more teaching goes ahead with better marks.

group C: is required additional only videos of the traditional teaching, so you can point out that the better marks are not only correlated to more teaching lessons.

introduction

qotation 7 contains the information that "grade point average..plays a role in success of hte dental hygiene students" the given information in the text is that this is the strongest indicator -> quotation error

in the text mentioned that GPA is a predictor, but quotation 7 states "..not necessarily indicate dental student success"

line 50 quotation error: Nuhat et al [10] again.

material and methods

line 93: students were randomly distributed into two groups. Which information is given to the students why they were divided in two groups -> this study is not blinded so the students should know that the CGPA is evaluated

line 101 the e-learning resources were selected by the authors which are the contributors to this course -> this study design works only if it is blinded -> the contributors of the traditional courses may influence the results

line 109 minimum of five views of one video of the e-learing platform, but what are the resources for the traditional group transcript of the contributors or only the own transcript 

line 116 to maintain the high level of objectivity two examiners evaluated the previous academic blocks. Why did you calculate an average score and not the Cohens Kappa? If this is an OSCE (objective structured clinical examination) there should be less differences if this score is objective

line 132 formation errors: duplicate blanks

line 133 no consistent notation of t test/t-test

results

the statistical analysis is weak: the preconditions for t-test or ANOVA is not calculated -> an independent t-test was conducted to evaluate the homogenety of group A and B? = should be Levene test

table 1 no consitent notation .002 vs table 2 0.163

table 3 bold font of the first row

line 161 no quotation to describe the values of the Pearson correlation in high, moderate and weak. It is definitely not Cohen (1988)

line 206 dental school in Germany

Reviewer 2 Report

Dear Authors, the topic is very interesting but I have some doubts. 

I think the rationale for this study needs to be made clearer. In particular, the connection between e-learning and the teaching of clinical skills in dental medicine could be made clearer. One way to demonstrate this connection would be to cite references (if possible) that show for example how in some highly industrialized countries the use of online lecture can was widely used during the Covid pandemic, this would make the article much more interesting. For Example I would suggest this papers :

Covid-19 pandemic and telephone triage before attending medical office: problem or opportunity? DOI10.3390/medicina56050250

In section Materials and Methods why don't you write the name of University ? I don't understand why !

Discussion- this paragraph should be rearranged. It is very chaotic. Please do not repeat information from Introduction and try to be more focused. Rewrite this section using following paragraphs: main results and clinical relevance; comparison with other studies; advantages and disadvantages of the study; conclusions and suggestions for future studies.

Taking everything in consideration, I strongly suggest that you rearrange the manuscript (especially Introduction and Discussion) and enlarge the sample size. Please, be more focused on the topic and do not repeat the same information several times.

I believe that your manuscript would have much more relevance after suggested improvements.

Reviewer 3 Report

Comments on Khattak et al:

The aim of this manuscript is to evaluate and compare the effectiveness of e – learning resources, as an additional teaching tool, for traditional teaching methods, in male and female students, based on the CGPA scores, in a pre-clinical operative skill course. An other significant aim of this manuscript is to correlate the effectiveness of e-learning resources, with CGPA scores of dental school students.

Even if the manuscript provides an organic overview, with a densely organized structure and based on well-synthetized evidence, there are aspects to be mentioned, to make the article fully readable. For these reasons, the manuscript requires minor changes.

Please find below an enumerated list of comments on my review of the manuscript:

INTRODUCTION:

LINE 32: New tecnologies play a pivotal role in medical and dental education, as they contribute in diversifying the teaching style, in the degree curricula (see, for reference: Varvara G, Bernardi S, Bianchi S, Sinjari B, Piattelli M. Dental Education Challenges during the COVID-19 Pandemic Period in Italy: Undergraduate Student Feedback, Future Perspectives, and the Needs of Teaching Strategies for Professional Development. Healthcare (Basel). 2021 Apr 12;9(4):454. doi: 10.3390/healthcare9040454. PMID: 33921516; PMCID: PMC8069889.) The authors should highlight the potential of this forefront learning tool, which has revolutionized the educational method, in recent years.

LINE 70 – 72: In this context, dental practice would benefit from introducing specialized training, which should include mentoring and extended practical training, as suggested by several and recent studies (see, for reference: Pinchi V, Varvara G, Pradella F, Focardi M, Donati MD, Norelli G. Analysis of professional malpractice claims in implant dentistry in Italy from insurance company technical reports, 2006 to 2010. Int J Oral Maxillofac Implants. 2014 Sep-Oct;29(5):1177-84. doi: 10.11607/jomi.3486. PMID: 25216146.).

MATERIAL AND METHODS:

As regards this section, the methodology design was rigorous and appropriately implemented within the study.

RESULTS and DISCUSSION:

Also these sections are well organized and densely presented, based on well-synthetized data.

In conclusion, this manuscript is densely presented and well organized, based on well-synthetized evidences. The authors were lucid in their style of writing, making it easy to read and understand the message, portrayed in the manuscript. Besides, the methodology design was rigorous and appropriately implemented within the study. However, many of the topics are very concisely covered. This manuscript provided a comprehensive review of current knowledge in this field. Moreover, this research have futuristic importance and could be potential for future research. However, I have minor comments only for the introductive section, for improvement before acceptance for publication. The article is accurate and provides relevant information on the topic and I suggest minor changes to be made in order to maximize its scientific impact. I would accept this manuscript, if the comments are addressed properly.

Round 2

Reviewer 2 Report

Dear Authors,

Thank you for revised manuscript